# Examining individual, health service, and experience of care determinants of patients' satisfaction - A cross-sectional facility-based study in KP, Pakistan

**Laurène Petitfour** [1,2*], **Stephan Brenner**[1], **Khalid Rehman**[3], **Swati Srivastava**[1], **Asad Ullah**[4], **Zohaib Khan**[4], **Manuela De Allegri**[1]

**1** Heidelberg Institute of Global Health, Medical Faculty and University Hospital, Heidelberg University, Im Neuenheimer Feld 130.3, Heidelberg, Baden-Württemberg 69120, Germany, **2** Aix Marseille University, Inserm, IRD, SESSTIM (Economic and Social Sciences of Health and Medical Information Processing), ISSPAM, 27 Bd Jean Moulin, Marseille, France **3** Institute of Public Health and Social sciences. Khyber Medical University. Peshawar. Pakistan, **4** Office of Research, Innovation, and Commercialization (ORIC), Peshawar, Khyber Pakhtunkhwa, Pakistan

\* laurene.petitfour@ird.fr

## Abstract

Health systems aim to meet populations' needs and to spur their satisfaction, requiring to first understand what drives this satisfaction, and whether these drivers come from the health system or not. In this empirical study, we use survey data from 666 patients exiting outpatient services of 60 public and 62 private health facilities in four districts of the province of Khyber Pakhtunkhwa, Pakistan. The survey tool assessed patients' socio-economic status, satisfaction with health service provision, and experiences of care (respect of privacy, out-of-pocket expenditures, availability of drugs). First, we synthesized the 16 items relating to satisfaction into a 12-item scale Patient Satisfaction score, relying on a Principal Component Analysis. Second, we used this score as the outcome in a linear model to identify its determinants. We found a generally high level of satisfaction regarding outpatient services within the sample, with an average satisfaction score of 76.37 out of 100. We did not find any association between satisfaction characteristics of the environment (type of facility, district). Women displayed significantly lower (−16%) and insured patients significantly higher (+6%) satisfaction scores. Indicators of healthcare experiences were found to be the strongest determinants of satisfaction, namely: satisfaction scores were 12% higher when privacy was respected, 5% higher when prescribed drugs were available and they also varied with the amount of healthcare expenses (positively and then negatively). Our results highlight the importance of designing policies that address the needs of specific population sub-groups and of ensuring access to core aspects of healthcare delivery, as reflected in patients' experiences, across all facility types.

**Data availability statement:** Data are available at https://doi.org/10.11588/DATA/VPDGXB.

**Funding:** Open Access funding enabled and organized by Projekt DEAL. We gratefully acknowledge financial support as part of a scientific collaboration with the KfW Development Bank ("Scientific implementation research on outpatient-department (OPD) services in the social health protection initiative, Khyber Pakhtunkhwa (KP) province and the Gilgit Baltistan (GB) area"). The funders had no role in study design, data collection and analysis, or preparation of the manuscript.

**Competing interests:** The authors have declared that no competing interests exist.

## Introduction

### Background

Health system responsiveness is defined as "how well a health system meets the legitimate expectations of the people it serves" [1,2] and is considered both as an objective *per se* and as an essential element in enabling Universal Health Coverage (UHC) [3]. As such, the role that health system responsiveness plays is central to the quality health service provision and the establishment of trust and respect between providers and their patients [4,5]. Responsive health services are more likely to ensure the recommended use of healthcare [6,7] and therefore more likely to result in better health outcomes [4], especially among chronically ill patients. Non-responsive healthcare provision, on the other hand, is more likely to contribute to poor treatment adherence among patients with chronic diseases [8].

Ideally, the assessment of health service responsiveness required information on how well the health system responds to legitimate patient expectations [9]. In the absence of information to scrutinize whether expectations are legitimate or not, empirical studies commonly rely on the assessment of patient experience or satisfaction, acknowledging that some expectations might be illegitimate or not under the control of the service provider. The assessment of patient experiences with receiving health care commonly focuses on perceptions related to the physical health care environment (e.g., amenities, cleanliness), service organization (e.g., waiting time), the patient-provider interactions (e.g., technical and interpersonal quality, opportunity to ask questions), or the outcome and affordability of care [10].

In comparison, the assessment of patient satisfaction is primarily focused on how patients rate different healthcare experiences [11,12]. Since patients' satisfaction with health care experiences are likely to be influenced by factors not directly related to the service provider or the overall health system, understanding associations between measured satisfaction and contextual factors, such as socio-demographic determinants of health (e.g., age, gender, education, socio-economic status, marital status, race, religion), or access to and use of care (geographic characteristics, visit regularity, health status) is critical [13,14].

In Pakistan, and specifically in the Khyber Pakhtunkhwa (KP) province, our study setting, the recent empirical evidence on patient satisfaction has provided some answers on the level of satisfaction in health facilities. Yet, several gaps remain. First, most studies have focused on tertiary facilities, reflecting satisfaction with inpatient care [15,16], rather than outpatient care [17]. This is in spite of the fact that from the health system perspective, strengthening the primary care level of care is crucial to ensure UHC [18]. Hence, it is crucial to understand patients' perspectives about their experience with primary level facilities and their satisfaction with care received at this level. Second, some studies have linked higher patient satisfaction to characteristics of patients [15–17] or their experience with receiving care, or to facility characteristics, but to our knowledge, none has considered both. In other Pakistani settings than KP, patient satisfaction in outpatient services was found to be associated with aspects of the patient-provider interaction, such as time to explain and answering

questions, ensuring privacy and respectfulness [19–21]. Individual characteristics like being female, uneducated, or employed were also found to be associated with higher satisfaction [21]. Other studies, conducted among diverse hospitals and tertiary care centres, considered facility characteristics and found that patients were generally more satisfied with the care received in the private sector compared to the public sector [22,23]. In synthesis, our review of existing literature identifies the need to conduct analyses of patients' satisfaction that account for multiple determinants simultaneously, generating more comprehensive evidence for policy.

Our study aims at identifying determinants of patients' satisfaction with outpatient care services at the primary care level in KP accounting for three sets of exposure variables: individual patient characteristics; health facility infrastructure; and patients' experience with health service delivery. Our study adds to the existing literature by examining patients' satisfaction in a remote region of the country, relying on a large sample of patients, drawn from a diverse sample of primary level facilities of various types, enabling us to include a more diverse set of exposure variables than prior studies.

## Methods

### Study setting

We collected data in four districts in the KP province of Pakistan: Chitral, Kohat, Malakand and Mardan. These districts were selected in agreement with the local health authorities as the target of an upcoming implementation project targeting primary healthcare services. The present study is embedded in the baseline survey of this project, so that no project activity had started at the time of data collection.

The healthcare delivery system in KP relies on both public and private facilities (see Fig 1). Public sector healthcare delivery follows a pyramidal structure: the bottom tier consists of Basic Health Units (BHU), which commonly include two to five medical practitioners that provide primary outpatient care for a modest fee of 10 PKR. The second tier consists of Rural Health Centers (RHCs), which can vary in both their size and their range of primary and some specialty services, as well as inpatient care. The top tiers include Tehsil Head Quarter (THQs) and District Head Quarter (DHQs) hospitals, which offer both inpatient and outpatient care for a range of specialty services. The private sector consists of individual or group-based general practitioners (GP) or specialists providing facility- or office-based care. Group-based private providers usually consist of two or more practitioners offering services at facilities managed by a single private authority. In both

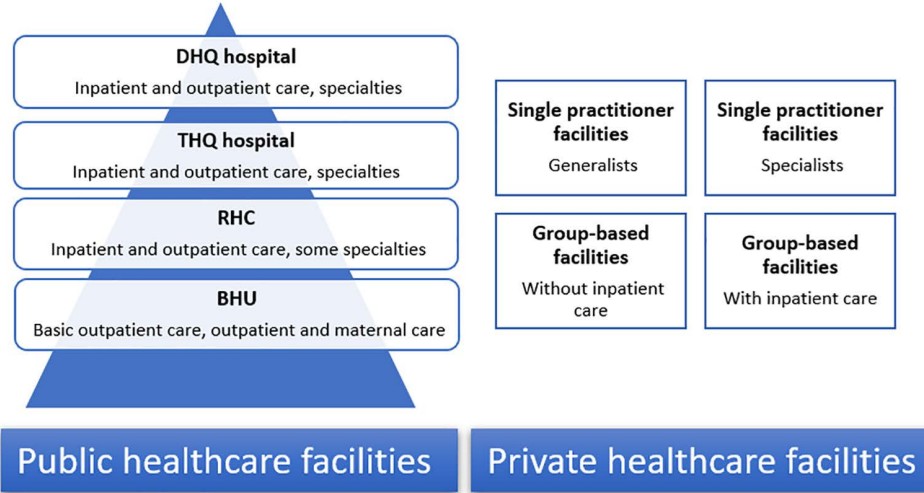

**Fig 1. Healthcare system in Khyber Pakhtunkhwa.**

private and public sector facilities direct user charges are the predominant modality, although these charges are highly subsidized in the latter.

## Sampling and interview setting

We used a two-stage cluster sampling strategy. We did not rely on sample size calculations, but rather aim at having enough patients in each type of facilities to identify individual and facility- level phenomenon. First, the number of facilities (our primary sampling unit) was defined to reflect the diversity of facilities offering primary outpatient services and remain feasible for field teams. The public sample of 60 facilities included a census of all DHQ and THQ hospitals, all RHCs, and 25 BHUs randomly selected from a total of 98 BHUs in the study area. We implemented a stratified selection of BHUs at the district level to ensure the district representation. The private sample included 62 facilities randomly selected from a list of 245 private facility provided to us by the KP Health Department. The 62 private facilities included 37 GPs (27 individual and 10 group-based practices), 11 clinical specialists (six individual, five group-based), and seven group-based outpatient service practices linked to an inpatient care facility. We implemented a stratified selection at the district level to ensure the district representation.

Second, we defined the number of patients (secondary sampling unit) to interview per sampled facility, aiming at capturing of individual situations and experiences, while remaining feasible within the time the surveyors could spend in each facility. We sat a minimum target of eight interviews per facility, but enumerators could interview more patients in larger facilities with greater patient attendance. We conducted exit interviews with patients exiting the facility premises after having attended outpatient care services. Our teams were stationed at each facility for one day. At each facility, on a given day, we included any adult patient exiting general outpatient services who consented to take part in the study, irrespective of their condition. The enumerators introduced themselves, and the study to the patients, and asked them for their consent to participate. They also asked a few questions to check their eligibility. Children or disabled patients that could not respond to the questionnaire on their own were excluded. Enumerators sat in a quiet and isolated location in or around the facility with consenting patients and administered the interview, lasting about 20 minutes. We collected data from September to November 2022 and from March to April 2023.

## Questionnaire

We collected information using a structured questionnaire, encompassing three sections. The first section collected information about a patient's socio-demographic characteristics and household assets, which we used to estimate socio-economic status. The second section included 20 questions collecting information on each patient's satisfaction with the healthcare they had just received. Within these 20 questions, the first four referred to patients' general satisfaction with the healthcare visit. Responses were captured on a 5-item Likert scale on: 1) the care-seeking experience; 2) the quality of care during this experience; 3) the quality of care in relation to the price; and 4) the likelihood to recommend this facility to others. The remaining 16 questions related to specific aspects of the healthcare encounter (e.g., respect shown by provider, respect of confidentiality). Nine out of these 16 questions also relied on a 5-item Likert scale to capture responses to questions phrased as: "How satisfied were you with <SPECIFIC HEALTHCARE ASPECT>?". Healthcare aspects included respect shown by provider, confidentiality level, capacity to explain diagnosis and potential treatments shown by provider, capacity to take time for questions shown by provider, capacity to take time shown by provider, waiting time, distance, adequacy of building, and adequacy of staff. All scaled responses were represented by five emoticons representing level of satisfaction ranging from "not smiling at all" to "much smiling" as depicted in Appendix 1, shown to the respondents to facilitate their understanding of the scale. The remaining seven questions elicited a simple binary response, YES vs NO, to the question "Were you satisfied with <SPECIFIC HEALTHCARE ASPECT>?". The concerned health aspects were cleanliness, waiting room, smell, access to clean toilet, access to clean water, access to soap, and disability access. The third section focused on aspects of patients' experience with the care received, which included questions about the

availability of medication at the facility and the occurrence and amount of any out-of-pocket expenditures (OOPE) for medical and non-medical items.

## Data analysis

Data analysis followed two stages. First, we derived a measure of patient satisfaction using information from the second questionnaire section. Next, we used this patient satisfaction metric as a dependent variable in a multivariate model examining the relation between satisfaction and the individual's socio-demographic profile and experience of care. The two stages are detailed below. We used Stata 15 to perform analyses.

## Construction of the composite satisfaction Index

During the first stage, we synthesized the information captured by all 20 questions related to satisfaction into one measure. Following the literature, we considered only aspects of satisfaction at this stage and did not yet consider actual elements of the healthcare experience [24]. We then computed the correlation matrix for the 20 satisfaction variables to check that they were all significantly and positively correlated. Next, we conducted a Principal Component Analysis (PCA) using only the 16 variables measuring satisfaction with specific dimensions of the healthcare experience (i.e., we excluded the four variables on general satisfaction). Indeed, we considered that introducing them in the composite index would blur the interpretation of the components, as the ultimate would include both general and specific variables. In addition, global satisfaction is likely to be mechanically correlated to the specific correlation items, as it is driven by them. Four variables were further removed because they exhibited more than 20% of missing values (clean water, clean toilet, disability access, soap access). After discussion with the field teams, these missing values were due to "non applicable" cases: for instance, many respondents mentioned not needing a specific access for disable people, or not needing toilets as it is only an outpatient clinic, hence they simply did not pay attention to their availability and did not answer the corresponding questions. Ultimately, the final PCA model included 12 of the initial 20 variables (see Table 1). The first component explained 41% of the total variability. We further tested the reliability of our measure and the correlation of each item with it by measuring the Alpha Cronbach score (=0.8525) and the Alpha score when removing each item one by one. We kept all items. We normalized the scores from this first component to range from 0 to 100 to ease interpretation. Hereafter, we refer to the derived composite variable as the Satisfaction Score.

Table 1. Variables included in the Satisfaction Score and characteristics of the first component.

| Eigenvalue of Component 1 = 4.99 | Proportion of variability explained by the first component | 41% |
|---|---|---|
| | Variables | Coef |
| How satisfied were you with the… | Respect shown by provider? | 0.34 |
| | Confidentiality level? | 0.30 |
| | Capacity to explain shown by provider? | 0.36 |
| | Capacity to take time for questions shown by provider? | 0.34 |
| | Capacity to take time shown by provider? | 0.38 |
| | Waiting time? | 0.30 |
| | Distance? | 0.26 |
| | Adequacy of building? | 0.28 |
| | Adequacy of staff? | 0.33 |
| Were you satisfied with the | Cleanliness? | 0.13 |
| | Waiting room? | 0.13 |
| | Smell? | 0.15 |

## Multivariate analysis of satisfaction score

During the second stage, we identified factors associated with patient satisfaction related to service delivery using a linear estimator to explain the composite score of satisfaction. We identified and selected exposure variables based on existing literature [13,25]. We introduced three types of exposure variables: variables related to the healthcare environment, variables related to patient characteristics, and variables related to a patient's experience with receiving care. All variables and their measurement are listed in Table 2. Our final model specification was the following:

$$Y_{i,j,k} = \alpha + \beta_k D + \gamma_j F + \delta_i X + \varepsilon_i$$

**Table 2. Exposure variables of the multivariate model.**

| Exposure variables | Type of variables: modalities | Expected sign and associated literature |
|---|---|---|
| **Environment** | | |
| District | 1 – Chitral<br>2 – Mardan<br>3 – Kohat<br>4 – Malakand | Unknown |
| Type of facility | 1 – BHU<br>2 – RHC<br>3 – Hospital<br>4 – Single practitioner private facility<br>5 – Group-based private facility | Higher satisfaction in private facilities [22,23] |
| **Individual characteristics** | | |
| Asset quintile | 1 – Lowest Asset Index quintile<br>2 – Second Lowest Asset Index quintile<br>3 – Middle Asset Index quintile<br>4 – Second Highest Asset Index quintile<br>5 – Highest Asset Index quintile | Higher satisfaction in richest quintiles [13,25] |
| Age | 1 - 18/25-year-old<br>2 - 26/39-year-old<br>3 - 40/54-year-old<br>4 - More than 55-year-old | Higher satisfaction with older age [13,25] |
| Gender | 0 – Male<br>1 – Female | Higher satisfaction for female [13,25] |
| BISP* beneficiary | 0 – No<br>1 – Yes | |
| Education | 1 – Completed Primary<br>2 – Completed Secondary, Completed university degree, Postgraduate<br>3 – No Formal Schooling or University Degree<br>4 – Other: Religious education (Madrasa), certificate | Higher satisfaction for better educated individuals [13,25] |
| Sehat Plus card owner | Categorical: Yes/No | Higher satisfaction for insured individuals |
| **Characteristics of the experience with healthcare seeking** | | |
| Was privacy maintained throughout your care? | 0 – No<br>1 – Yes | Higher satisfaction when privacy is respected [20,26] |
| Did you receive all the drugs you were prescribed? | 0 – No<br>1 – Yes | Higher satisfaction when drugs are available [27] |
| Amount of OOPE | Continuous (In transformed)<br><br>1 – No OOPE<br>2 – Between 1 and 10 PKR<br>3 – Between 11 and 800 PKR<br>4 – More than 800 PKR | Lower satisfaction when OOPE are high [28] |

* BISP = Benazir Income Support Programme

Where Y represents the Satisfaction Score of individual i in district k and facility j, $\alpha$ represents a constant, D represents a matrix of binary variables taking the value of 1 when an individual was surveyed in district k and 0 otherwise, F represents a matrix of binary variables taking the value of 1 when the individual was surveyed in a facility category j and 0 otherwise, and X represents a matrix of variables pertaining to characteristics and experiences of seeking healthcare of individual i. Finally, $\varepsilon_i$ represents the standard error term. We clustered the standard errors at the facility level, as the number of interviewed patients per facility varied. We checked the absence of collinearity between the exposure variables, as well as the normality of the residuals of the model.

Variables were introduced to the model following the empirical literature. The asset index was generated using Multiple Correspondence Analysis (MCA), building on the ownership of assets in the patient's household to estimate the socioeconomic status (SES) of the patient. The MCA included the ownership of refrigerator, washing machine, sofa, chair, cabinet, internet connection, bank account, and the composition of the roof and walls. For ease of interpretation, we categorized people into SES quintiles. We included a binary variable indicating whether a patient was covered by the *Sehat Plus Card*, an inpatient health insurance program, and a binary variable indicating whether a patient was a beneficiary of the *Benazir Income Support Programme* (BISP), a wide-scale cash transfer social protection program. Both programs target the poorest households. The rationale was to test the hypothesis that benefitting from a social protection program might increase familiarity and satisfaction with the local health system.

We also included three items capturing patients' experience of care, to remain focus on the key dimensions of service delivery, in line with prior literature [25] and with findings from our own qualitative work. This included privacy [20,26], drug availability [27], and the payment of out-of-pocket expenditures [28]. In our study setting, patient privacy was not always guaranteed due to inappropriate infrastructures or provider oversight, making patients potentially uncomfortable and dissatisfied. The unavailability of drugs has been a major challenge in primary healthcare facilities, both public and private, requiring patients to buy prescribed drugs from private pharmacies, thus incurring additional out-of-pocket expenditures and potentially causing healthcare foregoing. Lastly, we included the amount of medical out-of-pocket expenditures (OOPE) in logarithmic form to control for extreme values. We further tested potential non-linearity in the effect of OOPE introducing the square of OOPE in our model. We also implemented an alternative way to capture the potential non-linear effects of OOPE, using several categorical dummy variables: no OOPE, between 1 and 10 PKR, between 11 and 800 PKR and more than 800 PKR (all compared to the reference category of 1–10 PKR). Thresholds were chosen purposively to (i) isolate patients who did not incur any OOPE from the rest of patients, to (ii) distinguish between patients who paid the 10 PKR forfeit from patients paying higher OOPE, and to (iii) isolate expenses which represented about more than one day of GDP per capita at the time of the survey (annual GDP per capita in PKR/365.25). This threshold of 800 PKR allowed to isolate the highest quintile of OOPE.

### Ethics

This project received the approval of the Ethic committee of the Medical Faculty of the University of Heidelberg (S-492/2022) and received the ethics approval from Khyber Medical University ethics committee (KMU/EB/SI/00097). Written informed consent was obtained from each participant, and researchers had no access to information that could identify individual participants during or after data collection

### Results

#### Univariate statistics

Our sample consisted of 666 consenting respondents, of which 71% exited outpatient services of public facilities. The team could not reach the target of eight patients for all providers due to a low patient influx and not due to a refusal of consent. The respondent sample was quite balanced with regard to gender. Respondents with no education represented almost half of the sample (Table 3). Beneficiaries of the BISP social program represented nearly one quarter of the

sample, and almost one half of them (45%) were enrolled in the inpatient healthcare insurance (Sehat Plus Card). With respect to patient experiences with receiving healthcare, 79% of respondents reported that their privacy was protected, while only one half was able to receive all prescribed medicines directly from the provider. OOPE were unevenly distributed, with 12% of respondents encountering no OOPE and 45% paying a 10 PKR forfeit for their healthcare. The remaining respondents incurred OOPE ranging between 11 and 14900 PKR, with a higher density in lower values. Only 22% of respondents paid more than 800 PKR for their healthcare (Table 3). Patient satisfaction was overall high, with an average and median score of 76 out of 100.

## Bivariate statistics

Table 3 presents the mean satisfaction score for each exposure variable including the associated standard error and F-statistic corresponding to the equality of means across modalities tested. For all but two variables (i.e., facility type and age), we rejected the null hypothesis of equal means across modalities. The variables with the largest difference in mean satisfaction across modalities included gender, the Sehat Plus Card, and all variables related to the experience with healthcare.

## Multivariate analysis

Table 4 presents the results of the multivariate analysis with OOPE in logarithmic form (column a) and as categorical dummies (column b). Both models explained about 40% of patient satisfaction variability.

Among an individual's characteristics, gender and inpatient health insurance coverage displayed the strongest association with satisfaction. Being female decreased the satisfaction score by 16 points (p-value <0.01), while having a Sehat Plus Card was associated to a satisfaction score higher by about 6 points (p-value <0.01). The remaining individual characteristics did not substantially affect satisfaction, and neither did the type of facility.

Variables related to the experience of care had significant effect on patient satisfaction. Specifically, privacy protection during the care encounter increased the satisfaction score by 12 points (p-value <0.01), and receiving all prescribed drugs was associated with a significant increase by 5 points (p-value <0.01).

The association between OOPE and patient satisfaction exhibited a nonlinear relationship. In column (a) both the logarithm and the squared logarithm of the OOPE amount was significantly associated yet displayed contrary directions. The amount of OOPE was positively and then negatively associated with patient satisfaction (reversed U-curve). The turning point between the linear and the quadratic term was at 18.46 PKR and was consistent with the results presented in column (b). Compared with the reference group that paid 10 PKR, both patients who did not incur any OOPE (−8 percentage points, p-value<0.01) and those who paid between 11 and 800 PKR (−4 percentage points, p-value = 0.05) had lower satisfaction scores.

## Discussion

Relying on original survey data from 666 patients exiting the outpatient services of public and private providers, this study assessed the satisfaction of patients in relation to their experience with receiving care and examined factors associated with high satisfaction. Our findings indicated that satisfaction was generally high, but significantly lower for women. We found higher satisfaction rates associated with beneficiaries of the inpatient health insurance scheme available in the study area, with patients whose privacy had been protected during the care encounter, and with patients who had received all prescribed drugs. We also found that patients who paid small amounts of OOPE were more satisfied compared with both those not paying OOPE at all and with those paying greater amounts.

The high level of satisfaction detected is in line with the prior literature on patient satisfaction in Pakistan. Previous estimations of satisfaction were 64% in a sample of 380 geriatric patients in one tertiary public hospital in Karachi [29] and 87% in a sample of 1,095 diabetes patients exiting a public diabetes clinic in Lahore [21]. One study observed an increase

**Table 3. Distribution of exposure variables and bivariate statistics.**

| | Global statistics on satisfaction | | | | | |
|---|---|---|---|---|---|---|
| | Satisfaction score | No. | Mean | Std. deviation. | Median | |
| | Total | 666 | 76.37 | 19.61 | 77.36 | |
| | | No. | % | Mean Satisfaction score | Stand. err. of Satisfaction score | F stat (pvalue) |
| | **Environmental variables** | | | | | |
| District | Chitral | 93 | 14 | 72.07 | 2.12 | **7.57 (<0.001)** |
| | Mardan | 270 | 40.5 | 75.27 | 1.26 | |
| | Kohat | 148 | 22.2 | 74.59 | 1.61 | |
| | Malakand | 155 | 23.3 | 82.58 | 1.26 | |
| Type of facility | BHU | 100 | 15 | 74.81 | 2.01 | 1.77 (0.1337) |
| | RHC | 183 | 27.5 | 77.66 | 1.51 | |
| | Hospitals | 192 | 28.8 | 74.28 | 1.38 | |
| | Single practitioner private facility | 105 | 15.8 | 79.89 | 2.02 | |
| | Group-based private facility | 86 | 12.9 | 75.84 | 1.75 | |
| | **Individual variables** | | | | | |
| Asset quintile | 1 | 119 | 17.9 | 68.66 | 1.83 | **24.33 (<0.001)** |
| | 2 | 141 | 21.2 | 72.34 | 1.54 | |
| | 3 | 137 | 20.6 | 79.91 | 1.72 | |
| | 4 | 122 | 18.3 | 77.99 | 1.87 | |
| | 5 | 147 | 22.1 | 81.86 | 1.36 | |
| Age | 18/25-year-old | 149 | 22.4 | 73.71 | 1.65 | 1.88 (0.1309) |
| | 26/39-year-old | 268 | 40.2 | 77.99 | 1.20 | |
| | 40/54-year-old | 163 | 24.5 | 75.31 | 1.51 | |
| | More than 55-year-old | 86 | 12.9 | 77.97 | 2.03 | |
| Gender | Male | 340 | 51.1 | 85.83 | 0.85 | **213.18 (<0.001)** |
| | Female | 326 | 48.9 | 66.51 | 1.02 | |
| BISP beneficiary | No | 509 | 76.4 | 75.55 | 0.86 | **3.88 (0.0494)** |
| | Yes | 157 | 23.6 | 79.06 | 1.63 | |
| Sehat Plus card | No | 360 | 54.1 | 71.06 | 0.99 | **62.93 (<0.001)** |
| | Yes | 306 | 45.9 | 82.63 | 1.07 | |
| Education | Completed Primary | 137 | 20.6 | 82.69 | 1.60 | **11.03 (<0.001)** |
| | Completed Secondary / university degree | 193 | 29 | 77.99 | 1.29 | |
| | No Formal Schooling | 287 | 43.1 | 71.84 | 1.18 | |
| | Other: certificate, madrasa | 49 | 7.4 | 78.91 | 2.83 | |
| | **Aspects of the experience with seeking healthcare** | | | | | |
| Was privacy maintained throughout your care? | No | 139 | 20.9 | 69.71 | 2.01 | **20.89 (<0.001)** |
| | Yes | 527 | 79.1 | 78.13 | 0.78 | |
| Did you receive all the drugs you were prescribed? | No | 339 | 50.9 | 72.76 | 1.05 | **24.33 (<0.001)** |
| | Yes | 327 | 49.1 | 80.13 | 1.06 | |
| Amount of OOPE | No OOPE | 77 | 11.6 | 64.95 | 2.02 | **12.92 (<0.001)** |
| | Between 1 and 10 PKR | 310 | 46.7 | 79.54 | 1.10 | |
| | Between 11 and 800 PKR | 130 | 19.6 | 74.04 | 1.71 | |
| | More than 800 PKR | 147 | 22.1 | 77.75 | 1.56 | |

**Table 4. Determinants of the Satisfaction Index (linear model estimations).**

| | VARIABLES | Outcome: Satisfaction score (column a) *Coef (std. error)* | Outcome: Satisfaction score (column b) *Coef (std. error)* |
|---|---|---|---|
| **Environment** | **Districts (reference: Mardan)** | | |
| | Chitral | −4.52 (3.11) | −6.06 (3.40) * |
| | Kohat | −0.97 (2.31) | −2.00 (2.42) |
| | Malakand | 4.01* (2.38) | 3.58 (2.35) |
| | **Types of facilities (reference: BHU)** | | |
| | RHCs | 0.21 (3.10) | 0.28 (3.02) |
| | DHQ and THQ Hospitals | −4.06 (2.59) | −3.55 (2.60) |
| | Private single practitioner | −4.41 (3.78) | −5.33 (3.94) |
| | Private group-based facilities | −3.39 (4.04) | −6.09 (3.81) |
| **Individual characteristics** | **Gender (reference: male)** | | |
| | Female | −16.37 (2.14) *** | −16.59 (2.22) *** |
| | **Age (reference: 26/39-year-old)** | | |
| | 18/25-year-old | −3.26 (1.70) * | −2.86 (1.73) |
| | 40/54-year-old | --2.59 (1.41) * | −2.41 (1.43) * |
| | More than 55-year-old | −1.30 (2.24) | −1.08 (2.24) |
| | **Education (reference: no formal schooling)** | | |
| | Completed Primary | 2.09 (1.79) | 1.64 (1.82) |
| | Completed Secondary/University degree | −0.57 (1.74) | −1.21 (1.78) |
| | Diploma/certif. | 2.09 (2.62) | 1.60 (2.66) |
| | **Asset Index (reference: poorest quintile)** | | |
| | Asset Index quintile 2 | 2.10 (2.13) | 1.97 (2.13) |
| | Asset Index quintile 3 | 4.17* (2.52) | 3.87 (2.55) |
| | Asset Index quintile 4 | 2.49 (2.80) | 2.42 (2.76) |
| | Asset Index quintile 5 | 4.308* (2.60) | 4.21 (2.59) |
| | **Sehat Plus card beneficiary (reference: No)** | 6.65 (1.77) *** | 6.22 (1.81) *** |
| | **BISP BEN beneficiary (reference: No)** | 1.710 (1.90) | 1.63 (1.91) |
| **Experience with receiving healthcare** | **Privacy** | 11.82 (3.02)*** | 11.92 (3.06)*** |
| | **Received all prescribed drugs** | 5.03 (1.82) *** | 4.778 (1.85) ** |
| | **Out-of-pocket expenditures (reference:10 PKR)** | | |
| | No OOPE | | −8.13 (2.47)*** |
| | Between 11 and 800 PKR | | −4.18* (2.13) |
| | Above 800 PKR | | −3.72 (2.78) |
| | Ln (med expenses) | 4.02 (1.27)*** | |
| | Ln (med expenses) squared | −0.52 (0.17) *** | |

*(Continued)*

**Table 4.** (Continued)

| VARIABLES | Outcome: Satisfaction score (column a) *Coef (std. error)* | Outcome: Satisfaction score (column b) *Coef (std. error)* |
|---|---|---|
| Constant | 65.31 (4.90) *** | 73.79 (4.39) *** |
| Observations | 664 | 666 |
| R-squared | 0.412 | 0.407 |

Robust standard errors in parentheses, *** $p < 0.01$, ** $p < 0.05$, * $p < 0.1$

in the proportion of satisfied patients, exiting outpatient and gynaecological services, from 34% to 82% over the one-year period following the implementation of an intervention consisting of quality of care workshops sensitizing and training staff in more patient-centred interactions with women [30]. Other studies from Pakistan did not indicate the proportion of satisfied patients, but instead focused on relationships between patient satisfaction and patient expectation with the provider interaction [19,20,22,31].

This variability in findings is due to several reasons. First, the patient samples used across studies are not fully comparable in size and composition with respect to the healthcare providers (facilities of different or similar types) or the type of care studied (general, diabetes, geriatric). Samples and care-specific differences can explain gaps in gender, age and socio-economic characteristics that likely affect a patient's satisfaction with healthcare. Some samples include only female patients, while some focused on chronic diseases or elder. These profiles affect the estimated level of satisfaction, but with no mean of distinguishing them from other factors due to lack of variability. Second, measures used to estimate patient satisfaction are diverse in their content and definition. Studies using the SERVQUAL questionnaire, for instance, do not usually compute a synthetic measure of satisfaction but rather derive and compare satisfaction levels for each of five domains (reliability, responsiveness, assurance, empathy and tangibility) [30–32]. Other studies only assess patient's satisfaction with a more general level of care [29] or derive satisfaction measures based on interpersonal care aspects only [32].

By comparing patients across various provider types using several aspects of the healthcare experience, we were able to provide a more comprehensive assessment of patient satisfaction, which contributes and expands the currently available literature. Contrary to previous studies, we did not find that patients receiving care from private providers were more satisfied compared with public providers [22,23]. This might be explained by the inclusion of additional respondent characteristics and healthcare experiences in our model, which had a stronger effect on patient satisfaction compared with the type of facility alone.

The effect of gender on patient satisfaction has been contradictory in the literature. For instance, a review of determinants of patient satisfaction included seven studies that identified higher satisfaction levels among males and six studies that identified the contrary [13]. The relationship between patient gender and satisfaction is likely embedded in the local context and study setting and will require further investigations. For KP, the lower satisfaction of female patients might have been a reflection on the lack of female providers, both in public and private care settings, restricting provider choice for those women with a preference for female providers [33] and not corresponding to women's specific expectations.

While the positive impact of health insurance coverage on patient satisfaction has been observed previously [34], the significant and positive association between the Sehat Plus inpatient scheme and the satisfaction might seem surprising because patients did not benefit from any of its advantages in outpatient services. While we cannot explain it with certainty, it might be plausible that members of a social protection program are more familiar with or exposed to the healthcare system and value experiences in outpatient settings more favourably. More research, including qualitative studies, is needed to further understand the role of social protection schemes on patient experiences and satisfaction levels.

In our study, healthcare seeking experiences were crucial factors of patient satisfaction. For instance, privacy protection during the care process was positively associated with patient satisfaction. Indeed, patients who reported their privacy or confidentiality invaded, were more likely to refuse additional examinations or less likely to retain information shared by their provider, thus negatively impacting diagnostic accuracy or treatment efficacy [20]. Similarly, the availability of drugs was a major factor in determining positive patient satisfaction in our model, as drug shortages are a common occurrence in Pakistani healthcare facilities, and regularly cause treatment delays [35].

The reverse U-shape relationship between OOPE and satisfaction with a rather low turning point (18 PKR) confirmed previous results on the importance of affordability in the assessment of the healthcare experience by patients [28,36]. Patients seemed not to sanction low levels of OOPE in terms of satisfaction, but did so for amounts higher than 18 PKR. Given the negative impact of OOPE in terms of healthcare access and inequity reduction, the use and amount of user charges requires diligent attention and control to not contribute to the impoverishment or non-use of healthcare.

We acknowledge limitations in our study. First, some dimensions of patient experience were not captured by our questionnaire. For instance, we did not assess the reasons why patients sought care, such as the type or severity of their health condition. Hence, we might have omitted additional parameters in more comprehensively determining patients' expectations and experiences of healthcare that might have differently determined their satisfaction. Second, more precise information about the characteristics of the providers (their training, their personal attitudes, their motivation) with whom the patient interacted would enrich the analysis. Third, the size of our sample, the precise geographic setting and the non-purely random selection of patients within each facility might limit the overall generalizability of our findings to other settings. Fourth, the focus of our study is on outpatient care users, i.e., people who chose to seek care from the studied providers, which might have biased our satisfaction outcome to higher levels when compared with the population as a whole, since people with prior negative experiences of care might have been less likely to use the services again. Fifth, we recognise the weakness arising from the fact that in the absence of reliable health management information systems, we could not sample respondents to account for facility size. Hence, our analysis was not weighted for facility size, resulting in an overrepresentation of views from patients from smaller healthcare facilities. Given the emphasis on primary healthcare, this is not per se problematic. Lastly, our study bears the inherent limitations of patient satisfaction measures, meaning that it does not account for the legitimacy of a patient's expectations.

## Conclusion

Thanks to a diverse study sample, our empirical study offers a valuable contribution to the literature on the determinants of satisfaction and some actionable priorities to make the health system in KP more responsive to population needs and expectations. Satisfaction of patients exiting outpatient services is generally high, but with some discrepancies that allow us to identify areas to work on. First and foremost, respect for privacy, affordability and availability of appropriate treatment are essential to a positive experience with the healthcare system and should be monitored closely in all types of facilities. Additionally, the interpretation of the association between gender and satisfaction would require a deeper understanding of a patient's expectations and needs regarding the healthcare system, including per subpopulation (e.g., gender and age), mobilizing both researchers and policymakers. This additional layer would allow to make more informed decisions in terms of policy choices needed to build a more responsive system.

## Supporting information

**S1 Appendix. Emojis associated to each modality of the satisfaction questions.**
(DOCX)

**S2 Appendix. Inclusivity in global research questionnaire.**
(DOCX)

## Acknowledgments

We gratefully acknowledge the field team from Khybher Medical University for their contribution on the project, and the Kreditanstalt für Wiederaufbau (KfW) for its feedback on the paper.

## Author contributions

**Conceptualization:** Laurène Petitfour, Zohaib Khan, Manuela De Allegri.

**Data curation:** Laurène Petitfour.

**Formal analysis:** Laurène Petitfour, Stephan Brenner, Khalid Rehman, Manuela De Allegri.

**Funding acquisition:** Zohaib Khan, Manuela De Allegri.

**Investigation:** Asad Ullah, Zohaib Khan, Manuela De Allegri.

**Methodology:** Laurène Petitfour, Stephan Brenner.

**Project administration:** Khalid Rehman, Zohaib Khan, Manuela De Allegri.

**Resources:** Asad Ullah.

**Supervision:** Manuela De Allegri.

**Validation:** Stephan Brenner, Khalid Rehman, Swati Srivastava, Asad Ullah, Zohaib Khan, Manuela De Allegri.

**Writing – original draft:** Laurène Petitfour.

**Writing – review & editing:** Laurène Petitfour, Stephan Brenner, Khalid Rehman, Swati Srivastava, Asad Ullah, Zohaib Khan, Manuela De Allegri.

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
