## [Decision Letter · Decision Letter 0]

4 Jun 2025

PONE-D-25-05636What affects a patient’s satisfaction with healthcare? An empirical study from Patient Exit Interviews in Khyber Pakhtunkhwa, PakistanPLOS ONE

Dear Dr. Petitfour,

Thank you for submitting your manuscript to PLOS ONE. After careful consideration, we feel that it has merit but does not fully meet PLOS ONE’s publication criteria as it currently stands and requires a "major revision" before the journal can further consider this manuscript. Therefore, we invite you to submit a revised version of the manuscript that addresses the points raised during the review process. In addition, kindly provide all survey/study tools if you agree to resubmit the manuscript.

We look forward to receiving your revised manuscript.

Kind regards,

Dr. Syed Khurram Azmat, PhD, MPH, MD

Academic Editor

PLOS ONE

Journal Requirements:

[Open Access funding enabled and organized by Projekt DEAL. We gratefully acknowledge financial support as part of a scientific collaboration with the KfW Development Bank (“Scientific implementation research on outpatient-department (OPD) services in the social health protection initiative, Khyber Pakhtunkhwa (KP) province and the Gilgit Baltistan (GB) area”).].

4. Please expand the acronym “KfW” (as indicated in your financial disclosure) so that it states the name of your funders in full.

5. Thank you for stating the following in your manuscript:

[Open Access funding enabled and organized by Projekt DEAL. We gratefully acknowledge financial support as part of a scientific collaboration with the KfW Development Bank (“Scientific implementation research on outpatient-department (OPD) services in the social health protection initiative, Khyber Pakhtunkhwa (KP) province and the Gilgit Baltistan (GB) area”).]

[Open Access funding enabled and organized by Projekt DEAL. We gratefully acknowledge financial support as part of a scientific collaboration with the KfW Development Bank (“Scientific implementation research on outpatient-department (OPD) services in the social health protection initiative, Khyber Pakhtunkhwa (KP) province and the Gilgit Baltistan (GB) area”).]

7. Please amend the manuscript submission data (via Edit Submission) to include author Zohaid KHAN.

8. Please amend your authorship list in your manuscript file to include author Zohaib KHAN.

9. We note that Figure 1 in your submission contain map images which may be copyrighted. All PLOS content is published under the Creative Commons Attribution License (CC BY 4.0), which means that the manuscript, images, and Supporting Information files will be freely available online, and any third party is permitted to access, download, copy, distribute, and use these materials in any way, even commercially, with proper attribution. For these reasons, we cannot publish previously copyrighted maps or satellite images created using proprietary data, such as Google software (Google Maps, Street View, and Earth). For more information, see our copyright guidelines: http://journals.plos.org/plosone/s/licenses-and-copyright.

10. Please include captions for your Supporting Information files at the end of your manuscript, and update any in-text citations to match accordingly. Please see our Supporting Information guidelines for more information: http://journals.plos.org/plosone/s/supporting-information.

Reviewers' comments:

Reviewer's Responses to Questions

**Comments to the Author**

1. Is the manuscript technically sound, and do the data support the conclusions?

Reviewer #1: Yes

Reviewer #2: No

2. Has the statistical analysis been performed appropriately and rigorously? 

Reviewer #1: Yes

Reviewer #2: No

3. Have the authors made all data underlying the findings in their manuscript fully available?

Reviewer #1: Yes

Reviewer #2: No

4. Is the manuscript presented in an intelligible fashion and written in standard English?

Reviewer #1: Yes

Reviewer #2: Yes

5. Review Comments to the Author

Reviewer #1: Dear Authors,

Congratulations on the successful submission of your manuscript titled . I commend you for your hard work and dedication in bringing this research to fruition.

With minor edits and clarifications, this paper is ready for publication. The research is original, well-designed, and has strong potential policy relevance for healthcare delivery in LMIC contexts.

A general comment is to use smaller sentences; some sentences, especially in introduction and discussion sections are very long, these can be split into smaller sentences for better readability. Please see additional comments/suggestions in the attached document and address them for resubmission.

Best of luck and warm regards,

Reviewer #2: It is a good study that attempted to study determinants of patient satisfaction in the Khyber Pakhtunkhwa province of Pakistan. However, it suffer some major limitations and ambiguities. The detailed feedback is presented below

Title:

"What affects a patient’s satisfaction with healthcare?"

• The title is currently too broad. Consider specifying the determinants being assessed — for example, whether they are related to health system factors, patient characteristics, or both. The current phrasing is vague and lacks focus.

Abstract:

• Clearly state the objective of the study at the outset.

• Specify the tool or instrument used to assess satisfaction (e.g., “A 13-item scale was used to measure patient satisfaction…”).

• To improve clarity, organize the results by factor type — for instance, patient characteristics versus systemic or facility-level factors. Currently, these are mixed together.

• The sentence “being insured by an inpatient-focused insurance was significantly associated with satisfaction” requires clarification. Since the survey targeted patients exiting outpatient services, it's unclear how inpatient insurance coverage applies here.

Data Availability:

• The statements are contradictory: first, it says “No – some restrictions will apply”, then it claims data will be publicly available. Please clarify the actual data availability status and correct inconsistencies.

Introduction:

• Include studies conducted in Khyber Pakhtunkhwa (KP) province to enhance contextual relevance. Consider citing other papers:

o https://journalofkcd.com/kcd/article/view/25

o https://jpma.org.pk/index.php/public_html/article/view/9990

• The rationale for the study is currently weak. Clearly highlight specific gaps in the existing literature and explain how this study addresses them. The current explanation is generic and does not clearly demonstrate the study’s unique contribution.

• The phrase “…identifying determinants of patients’ satisfaction…” should be more specific. Explicitly mention whether these determinants include demographic, financial, structural, or service-related factors.

Methods:

• Clearly state the study design and the rationale for the selection of districts. Was the sampling random, purposive, or part of a larger study?

• Define the eligibility criteria for participants — such as age, type of medical condition, or type of healthcare service utilized.

• Explain how the sample size was determined. If this is a secondary analysis, indicate whether the existing sample was sufficient for your study objectives.

• Describe the interview setting, duration, and who conducted the interviews.

• Line 119–120: Clarify whether these variables were additional to the 21 satisfaction items or overlapping.

• Line 124–125: The phrase “capacity to explain…” is unclear and should be reworded for precision. Similarly, ‘rate of the quality of care’?

• Line 126–127: “It further included eight binary satisfaction variables…” needs clarification. If these overlap with the previously mentioned variables, reorganize and integrate this information earlier for coherence.

Construction of composite index

• Line 144: Explain why four general satisfaction variables were excluded.

• Line 145: Explain the reason for removal of variables with clear statistical criteria (e.g., low factor loadings or communalities). Mention the cut-off values used.

• Line 148: Clarify the rationale for using normalized eigenvalues as satisfaction scores instead of first principal component. The eigenvalues tell us the variance component of PCA.

• Overall, the section on satisfaction item lack conceptual clarity. Whether the tool was validated or whether the 17 satisfaction items comprehensively captures all the dimensions of the satisfaction construct. A conceptual framework or mapping of domains would enhance clarity.

Statistical Analysis:

• Patients were sampled equally across facilities, despite variations in patient volume. Explain why weighting was not used to adjust for this imbalance.

Table 2:

• The stated rationale “Higher satisfaction in private facilities” needs clarification. It does not align clearly with the indicator ‘asset quintile’.

• For the Sehat Plus Card variable:

o a) Re-confirm whether all residents of KP were eligible at the time of data collection or if it was limited to low-income groups. Per my understanding, SCP services for everyone in the KP who has a KP national identity card.

o b) Since the Sehat Card covers only inpatient services, please explain the relevance of this indicator when the focus is on outpatient care.

Line 184:

• The dimension of ‘experience with healthcare seeking’ doesn’t seem to be studied comprehensively. It merely has only 3 indicators.

Line 192–194:

• Strengthen the justification for cut-off used for out-of-pocket expenditure (OOPE).

• Provide more detail on the modeling approach: “We tested our model for both the logarithmic of OOPE and its square”.

Results:

• In Table 3, clarify what the values in parentheses represent — are these standard errors, or something else?

Discussion:

• Not accounting for the type of service received by patient is also a limitation.

Overall feedback:

While some of the findings are interesting, the study suffer major limitations: a) the rationale is weak as it does not specifically highlight the gaps in existing literature on the topic and how this particularly study addresses these gaps and add unique value to the existing body of knowledge; b) the study revolves around patient satisfaction; however, the assessment of satisfaction lack conceptual underpinnings. Furthermore, the validity and reliability of the tool and the way construct is developed is unclear. Factor analysis would have been a better technique to develop the overall satisfaction construct; c) certain dimensions of determinants are not comprehensively captures such as environmental and experience of health seeking only has 2 and 3 variables, respectively.

6. PLOS authors have the option to publish the peer review history of their article (what does this mean?). If published, this will include your full peer review and any attached files.

Reviewer #1: **Yes:** SYED FARHAN ALI TIRMIZI

Reviewer #2: No

---

## [Author Response · Author response to Decision Letter 1]

9 Nov 2025

Dear editor,

we provided in the uploaded document a detailed rebuttal letter

Best regards,

Laurène Petitfour

---

## [Decision Letter · Decision Letter 1]

30 Dec 2025

PONE-D-25-05636R1Examining individual, health service, and experience of care determinants of patients’ satisfaction - a cross-sectional facility-based study in KP, PakistanPLOS One

Dear Dr. Petitfour,

Thank you for submitting your manuscript to PLOS ONE. After careful consideration, we feel that it has merit but does not fully meet PLOS ONE’s publication criteria as it currently stands. Therefore, we invite you to submit a revised version of the manuscript that addresses the points raised during the review process. A minor revision is required before the journal can consider the manuscript for publication.

We look forward to receiving your revised manuscript.

Kind regards,

**Dr Syed Khurram Azmat**, PhD, MPH, MD

Academic Editor

PLOS One

Journal Requirements:

Reviewers' comments:

Reviewer's Responses to Questions

**Comments to the Author**

1. If the authors have adequately addressed your comments raised in a previous round of review and you feel that this manuscript is now acceptable for publication, you may indicate that here to bypass the “Comments to the Author” section, enter your conflict of interest statement in the “Confidential to Editor” section, and submit your "Accept" recommendation.

Reviewer #1: All comments have been addressed

2. Is the manuscript technically sound, and do the data support the conclusions?

Reviewer #1: Partly

3. Has the statistical analysis been performed appropriately and rigorously? 

Reviewer #1: Yes

4. Have the authors made all data underlying the findings in their manuscript fully available?

Reviewer #1: No

5. Is the manuscript presented in an intelligible fashion and written in standard English?

Reviewer #1: Yes

6. Review Comments to the Author

Reviewer #1: General Assessment

The manuscript tackles a very relevant, important topic. The data is unique, coming from several districts, and encompasses public, as well as private, facilities. The methodological toolkit is, overall, appropriate, and the manuscript is well-explained. However, there appear several aspects, mainly related to sampling and sample size, that need clarification, concerning the overall goal of the study, with regard to intrinsic coherence. These points do not make the manuscript incorrect, per se, but need correction.

MAJOR COMMENTS

1. Absence of Sample Size Justification: The manuscript does not provide a sample size calculation or justification, either at the facility or patient level.

Suggested action: Explicitly state whether the sample size was determined by feasibility/programmatic constraints, or a formal power calculation was conducted. If no formal calculation was done, authors should justify why the achieved sample (n=666) is sufficient for multivariable modeling, and acknowledge this as a limitation.

2. Unclear Primary Sampling Unit (Facility vs. Patient): The manuscript does not clearly define the sampling unit, and the description alternates between facilities as the unit of selection, and patients as the unit of analysis. This ambiguity affects interpretation of representativeness, variance estimation, and external validity.

Suggested action: Clearly state the primary sampling unit (PSU), the secondary sampling unit, and how this informed the analytical approach.

3. Sampling Strategy: The described approach resembles a two-stage cluster sampling design, but this is never explicitly stated, nor analytically accounted for beyond clustering SEs.

Suggested action: Authors should explicitly describe the design as facility-based cluster sampling, acknowledge that patient selection was non-random, and discuss implications for generalizability.

4. Inconsistency ‘8 Interviews per Provider’ vs ‘per Facility’: There is inconsistency in the Methods section. In some places, the target is described as 8 interviews per provider; elsewhere, it is described as 8 interviews per facility. This is confusing, particularly given that facilities may include multiple providers.

Suggested action: Standardize terminology throughout the manuscript. Clarify whether the target was per facility-day, or per individual provider within facilities.

5. Shift in Study Objective (Satisfaction Determinants vs. Health Insurance)

The stated objective focuses on “…individual characteristics; health facility infrastructure; and patients’ experience with health service delivery.” However, health insurance (Sehat Plus Card) appears prominently in the analysis and discussion. The framing increasingly suggests insurance as a key exposure, despite outpatient services not being covered. This creates conceptual drift between stated objectives, analytical emphasis, and interpretation of findings.

Suggested action: Either explicitly include insurance/social protection as a determinant in the stated objectives, or reframe insurance as an exploratory or secondary exposure.

MINOR COMMENTS

Discussion

• Interpret the insurance finding more cautiously, emphasizing that causality cannot be inferred.

Language & Style

• Minor grammatical edits needed (e.g., tense consistency, phrasing such as “abidance” vs. “adherence”).

• Some sentences are overly long and would benefit from simplification.

7. PLOS authors have the option to publish the peer review history of their article (what does this mean?). If published, this will include your full peer review and any attached files.

Reviewer #1: **Yes:** SYED FARHAN ALI TIRMIZI

---

## [Author Response · Author response to Decision Letter 2]

26 Feb 2026

We thank the reviewer for their careful reading of our manuscript, and provide answers to the remaining comments one by one. All of our answers can be found in the "Rebuttal letter" document uploaded on your server.

Reviewer #1: General Assessment

MAJOR COMMENTS

1. Absence of Sample Size Justification: The manuscript does not provide a sample size calculation or justification, either at the facility or patient level.

Suggested action: Explicitly state whether the sample size was determined by feasibility/programmatic constraints, or a formal power calculation was conducted. If no formal calculation was done, authors should justify why the achieved sample (n=666) is sufficient for multivariable modelling, and acknowledge this as a limitation.

We thank you for your comment, and your careful reading of the method section. We worked on the Study setting and the Sampling and interview setting subsections to make our sampling clearer and more explicit. Indeed, we did not rely on formal sample size calculations.

2. Unclear Primary Sampling Unit (Facility vs. Patient): The manuscript does not clearly define the sampling unit, and the description alternates between facilities as the unit of selection, and patients as the unit of analysis. This ambiguity affects interpretation of representativeness, variance estimation, and external validity.

Suggested action: Clearly state the primary sampling unit (PSU), the secondary sampling unit, and how this informed the analytical approach.

We took your recommendation and explained in the text the primary sampling unit (the facility) and the secondary sampling unit (the patient).

We present at the end of the 5 major comments the former and new version of these subsections.

3. Sampling Strategy: The described approach resembles a two-stage cluster sampling design, but this is never explicitly stated, nor analytically accounted for beyond clustering SEs.

Suggested action: Authors should explicitly describe the design as facility-based cluster sampling, acknowledge that patient selection was non-random, and discuss implications for generalizability.

We explained in the text the two-stage clustering sampling strategy, and added a sentence in the limitations. Selection of patient within facilities was indeed not random, because the number of patients per day was not sufficient in many facilities to implement any consistent and random selection strategy.

4. Inconsistency ‘8 Interviews per Provider’ vs ‘per Facility’: There is inconsistency in the Methods section. In some places, the target is described as 8 interviews per provider; elsewhere, it is described as 8 interviews per facility. This is confusing, particularly given that facilities may include multiple providers.

Suggested action: Standardize terminology throughout the manuscript. Clarify whether the target was per facility-day, or per individual provider within facilities.

We amended the terminology where it needed to be, and checked that it is consistent throughout the text.

5. Shift in Study Objective (Satisfaction Determinants vs. Health Insurance)

The stated objective focuses on “…individual characteristics; health facility infrastructure; and patients’ experience with health service delivery.” However, health insurance (Sehat Plus Card) appears prominently in the analysis and discussion. The framing increasingly suggests insurance as a key exposure, despite outpatient services not being covered. This creates conceptual drift between stated objectives, analytical emphasis, and interpretation of findings.

Suggested action: Either explicitly include insurance/social protection as a determinant in the stated objectives, or reframe insurance as an exploratory or secondary exposure.

We thank you for this comment which helped us improve the clarity of the manuscript. Indeed, this specific study is embedded in a larger implementation research project, which addresses the roll-out of a health insurance project. The four study districts were chosen for the purpose of this project and the focus on primary outpatient services was also derived from the research objective of this project.

On the contrary, the sampling of the facilities and of patients within facilities were done for the purpose of this specific study.

We initially thought bringing information about the broader project would be helpful, but in light of your comments, we think it creates more confusion, and removed precise reference to the project and its content.

---

## [Editor Report · Decision Letter 2]

24 Mar 2026

Examining individual, health service, and experience of care determinants of patients’ satisfaction - a cross-sectional facility-based study in KP, Pakistan

PONE-D-25-05636R2

Dear Dr. Petitfour,

We’re pleased to inform you that your manuscript has been judged scientifically suitable for publication and will be formally accepted for publication once it meets all outstanding technical requirements.

Kind regards,

Syed Khurram Azmat, PhD, MPH, MD

Academic Editor

PLOS One
---

## [Editor Report · Acceptance letter]

PONE-D-25-05636R2

PLOS One

Dear Dr. Petitfour,

I'm pleased to inform you that your manuscript has been deemed suitable for publication in PLOS One. Congratulations! Your manuscript is now being handed over to our production team.

Kind regards,

on behalf of

Dr. Syed Khurram Azmat

Academic Editor

PLOS One